# EMERGENCE OF A HIGH-DIMENSIONAL ABSTRACTION PHASE IN LANGUAGE TRANSFORMERS

**Emily Cheng**[1], **Diego Doimo**[2], **Corentin Kervadec**[1], **Iuri Macocco**[1], **Jade Yu**[3]
**Alessandro Laio**[4], **Marco Baroni**[1,5]
Universitat Pompeu Fabra[1], Area Science Park[2], University of Toronto[3], SISSA[4], ICREA[5]
`emilyshana.cheng@upf.edu`

## ABSTRACT

A language model (LM) is a mapping from a linguistic context to an output token. However, much remains to be known about this mapping, including how its geometric properties relate to its function. We take a high-level geometric approach to its analysis, observing, across five pre-trained transformer-based LMs and three input datasets, a distinct phase characterized by high intrinsic dimensionality. During this phase, representations (1) correspond to the first full linguistic abstraction of the input; (2) are the first to viably transfer to downstream tasks; (3) predict each other across different LMs. Moreover, we find that an earlier onset of the phase strongly predicts better language modelling performance. In short, our results suggest that a central high-dimensionality phase underlies core linguistic processing in many common LM architectures.

 https://github.com/chengemily1/id-llm-abstraction

## 1 INTRODUCTION

Compression is thought to underlie generalizable representation learning (Deletang et al., 2024; Yu et al., 2023). Indeed, language models compress their input data to a manifold of dimension orders-of-magnitude lower than their embedding dimension (Cai et al., 2021; Cheng et al., 2023; Valeriani et al., 2023). Still, the *intrinsic dimension* (ID) of input representations may fluctuate over the course of processing. We ask: what does the evolution of ID over layers reveal about representational generalizability, and, broadly, about linguistic processing?

The layers of an autoregressive language model (LM) transform the LM's input into information useful to predict the next token. In this paper, we characterize the geometric shape of this transformation across layers, uncovering a profile that generalizes across models and inputs: **(1)** there emerges a distinct phase characterized by a peak in the intrinsic dimension of representations; **(2)** this peak is significantly reduced in presence of random text and nonexistent in untrained models; **(3)** the layer at which it appears correlates with LM quality; **(4)** the highest-dimensional representations of different networks predict each other, but, remarkably, neither the initial representation of the input nor representations in later layers; **(5)** the peak in dimension marks an approximate borderline between representations that perform poorly and fairly in syntactic and semantic probing tasks, as well as in transfer to downstream NLP tasks.

Taken together, our experiments suggest that all analyzed transformer architectures develop, in the intermediate layers, a high-dimensional representation encoding abstract linguistic information. The results of this processing are stored in representations which are then used, possibly through a process of incremental refinement, to predict the next token. Besides providing new insights on the inner workings of Transformer-based LMs, these results have implications for tasks such as layer-based fine-tuning, model pruning and model stitching.

## 2 RELATED WORK

The remarkable performance of modern LMs, combined with the opacity of their inner workings, has spurred a wealth of research on interpretability. At one extreme, there are studies that benchmark

LMs treated as black boxes (e.g., Liang et al. (2023)). At the other extreme, researchers are "opening the box" to mechanistically characterize how LMs perform specific tasks (e.g.,Meng et al. (2022); Conmy et al. (2023); Geva et al. (2023); Ferrando et al. (2024)). We take the middle ground, using geometric tools to characterize the high-level activation profiles of LMs, and we relate these profiles to their processing behaviour. In particular, we draw inspiration from the *manifold hypothesis*, or the idea that real-life high-dimensional data often lie on a low-dimensional manifold (Goodfellow et al., 2016): we estimate the *intrinsic dimension* of the representational manifold at each LM layer to gain insight into how precisely layer geometry relates to layer function.

The notion that nominally complex, high-dimensional objects can be described using few degrees of freedom is not new: it underlies, for instance, a number of popular dimensionality reduction methods such as standard (linear) PCA (Jolliffe, 1986). But, while PCA is linear, the data manifold need not be: as such, general nonlinear methods have been proposed to estimate the *topological dimension*, or manifold dimension, of point clouds (see Campadelli et al. (2015) for a survey). As neural representations tend to constitute nonlinear manifolds across modalities (Ansuini et al., 2019; Cai et al., 2021), we use a state-of-the-art nonlinear ID estimation method, the Generalized Ratios Intrinsic Dimension Estimator (GRIDE) (Denti et al., 2022), which we further describe in Section 3.4.

Deep learning problems tend to be high-dimensional. But, recent work reveals that these ostensibly high-dimensional problems are governed by low-dimensional structure. It has, for instance, been shown that common learning objectives and natural image data lie on low-dimensional manifolds (Li et al., 2018; Pope et al., 2021; Psenka et al., 2024); that learning occurs in low-dimensional parameter subspaces (Aghajanyan et al., 2021; Zhang et al., 2023); that modern neural networks learn highly compressed representations of images, protein structure, and language (Ansuini et al., 2019; Valeriani et al., 2023; Cai et al., 2021); and, moreover, that lower-ID tasks and datasets are easier to learn (Pope et al., 2021; Cheng et al., 2023).

In the linguistic domain, considerable attention has been devoted to the ID of LM *parameters*. Zhang et al. (2023) showed that task adaptation occurs in low-dimensional parameter subspaces, and Aghajanyan et al. (2021) that low parameter ID facilitates fine-tuning. In turn, the low effective dimensionality of parameter space motivates parameter-efficient fine-tuning methods such as LoRA (Hu et al., 2022), which adapts pre-trained transformers using low-rank weight matrices.

Complementary to parameter ID, a number of works focus on the ID of *representations* in LM activation space. In particular, Cai et al. (2021) were the first to identify low-dimensional manifolds in the contextual embedding space of (masked) LMs. Balestriero et al. (2023) linked representational ID to the scope of attention and showed how toxicity attacks can exploit this relationship. Tulchinskii et al. (2023) demonstrated that representational ID can be used to differentiate human- and AI-generated texts. Cheng et al. (2023) established a relation between representational ID and information-theoretic compression, also showing that dataset-specific ID correlates with ease of fine-tuning. Yin et al. (2024) showed that the local ID of specific inputs in specific layers can be used to diagnose model truthfulness.

Closer to our aims, Valeriani et al. (2023) studied the evolution of ID across layers for vision and protein transformers, with a preliminary analysis of a single language transformer tested on a single dataset (a similarly preliminary analysis is also provided by (Yin et al., 2024)). Like us, Valeriani and colleagues found that ID develops in different phases, with a consistent early peak followed by a valley and less consistent markers of a second peak. We greatly extend their analysis of linguistic transformers by investigating the functional role of the main ID peak in five distinct LMs. Importantly, our converging evidence suggests that, in language transformers, semantic processing of the input first takes place under the early ID peak, *contra* the preliminary evidence by Valeriani and colleagues that this crucial phase is the low-ID "valley" between the peaks.

## 3 METHODS

### 3.1 MODELS

We consider five causal LMs of different families, namely OPT-6.7B (Zhang et al., 2022), Llama-3-8B (Meta, 2024), Pythia-6.9B (Biderman et al., 2023), OLMo-7B Groeneveld et al. (2024), and

Mistral-7B (non-instruction-tuned) (Jiang et al., 2023), hereon referred to as OPT, Llama, Pythia, OLMo, and Mistral, respectively.

OPT, Pythia, and OLMo make public their pre-training datasets, which are a combination of web-scraped text, code, online forums such as Reddit, books, research papers, and encyclopedic text (see Zhang et al. (2022); Gao et al. (2020); Soldaini et al. (2024), respectively, for details). The pre-training datasets of Llama-3 and Mistral are likely similar, though they remain undisclosed at the time of publication.

All language models considered have between 6.5 and 8B parameters, 32 hidden layers, and a hidden dimension of 4096. Moreover, they all inherit the architectural design of the decoder-only transformer (Vaswani et al., 2017) with a few notable changes: 1) layer normalization operations are applied before self-attention and MLP sublayers; 2) Pythia adds the self-attention and MLP sublayer outputs to the residual stream in parallel, while in other models self-attention outputs are added to the residual stream before the MLP; 3) Llama/Mistral/OLMo replace the ReLU activation function with SwiGLU (Shazeer, 2020), and Pythia uses GeLU instead (Hendrycks & Gimpel, 2016); 4) all models but OPT replace absolute positional embeddings with rotary positional embeddings (Su et al., 2022); 5) to facilitate self-attention computation, Llama uses grouped query attention (Ainslie et al., 2023), Mistral applies a sliding window attention, and Pythia adopts Flash Attention (Dao et al., 2022). In all cases, we use as our per-layer representations the vectors stored in the HuggingFace transformers library `hidden_states` variable.[1]

In autoregressive models, due to the causal nature of attention, an input sequence's last token representation is the only one to contain information about the whole sequence. Furthermore, it is the only one that is decoded at the last layer to predict the next token. For these reasons, we choose to represent input sequences with their *last token representation* at each layer.

## 3.2 DATA

Since we focus on model behavior in-distribution, we compute observables using three corpora that proxy models' pre-training data (all accessed through HuggingFace): Bookcorpus (Zhu et al., 2015); the Pile (Gao et al. (2020); precisely, the 10k-document subsample available on HuggingFace) and WikiText-103 (Merity et al., 2017). From each corpus, we sample, without replacement, a total of 50k distinct 20-token sequences (sequence length is counted according to the number of typographic tokens in the text of each corpus).[2] We then divide these samples into partitions of 10k sequences each, which we use for the experiments. We do not constrain the sequences to have any particular structure: in particular, their final element is not required to coincide with the end of a sentence. We additionally generate 5 partitions of 10k 20-token sequences from *shuffled* versions of the same corpora. That is, we first randomize the order of the tokens in each corpus, and then proceed as with the non-randomized versions. The shuffled samples respect the source corpus unigram frequency distribution, but syntactic structure and semantic coherence are destroyed.

## 3.3 PROBING AND DOWNSTREAM TASKS

To relate representations' geometry to their content, we use the probing datasets of Conneau et al. (2018), meant to capture the encoding of surface-related, syntactic, and semantic information in LM representations. For each layer, we train a lightweight MLP probe from the hidden representation to each linguistic task. Details on tasks and training procedure are given in Appendix I.

If there is a relation between ID and abstract linguistic processing, then ID may also predict ease-of-transfer to downstream NLP tasks. To test this claim, we consider two such tasks, sentiment classification of film reviews (Maas et al., 2011) and toxicity classification (Adams et al., 2017), for which we train binary linear classifiers on each hidden layer representation (training details in Appendix J). Note that we do not fine-tune the layers themselves, as this might change their IDs, which is the very property we want to relate to task performance.

---

[1]Each `hidden_state` vector corresponds to the representation in the *residual stream* (Elhage et al., 2021) after one attention and one MLP update.

[2]We replicated the Pile-based Pythia and OPT experiments with sequences extended to 128 tokens. We obtained very similar results, confirming that the sequences we are using cover the typical contextual spans encoded in model representations.

## 3.4 INTRINSIC DIMENSION

Real-world datasets tend to show a high degree of possibly non-linear correlations and constraints between their features (Tenenbaum et al., 2000). This means that, despite a very large embedding dimension, data typically lie on a (locally smooth) manifold characterized by a much lower dimensionality, referred to as its intrinsic dimension (ID). This quantity may be thought of as the number of independent features needed to locally describe the data with minimal information loss (Bishop, 1995). Equivalently, it has been defined as the dimensionality of the support of the probability distribution from which the data is generated (Fukunaga, 2013; Campadelli et al., 2015).

In almost every real-world system, the ID depends on the scale, i.e., the size of neighbourhood at which the data is analyzed. In particular, at small scales, the true dimensionality of the manifold is typically hidden by that of data noise. At very large scales, the ID estimate can also be erroneous, due, for instance, to the curvature of the manifold (Facco et al., 2017; Denti et al., 2022). For this reason, in order to obtain a reliable and meaningful ID estimation, a proper scale analysis is necessary. This is typically performed by varying the amount of the neighbours considered in estimating the ID and looking for an interval of scales in which the estimate is approximately stable (see Appendix C for further details). To this aim, we opted for the *generalized ratios intrinsic dimension estimator* (GRIDE) of Denti et al. (2022), which extends the commonly used[3] TwoNN estimator of Facco et al. (2017) to general scales. We selected GRIDE because it allows probing, in a rigorous framework, the dependence of ID on scale. While the original TwoNN estimator assumes local uniformity up to the 2nd nearest neighbor, GRIDE relaxes this assumption to produce unbiased ID estimates up to the $2k$th nearest neighbor ($k$ being the scale).

In GRIDE, the fundamental ingredients are ratios $\mu_{i,2k,k} = r_{i,2k}/r_{i,k}$, where $r_{i,j}$ is the Euclidean distance between point $i$ and its $j$-th nearest neighbour. Under local uniform density assumptions, the $\mu_{i,2k,k}$ follow a generalised Pareto distribution $f_{\mu_{i,2k,k}}(\mu) = \frac{d(\mu^d-1)^{k-1}}{B(k,k)\mu^{d(2k-1)+1}}$ that depends on the intrinsic dimension $d$ of the manifold, where $B(\cdot,\cdot)$ is the beta function. By assuming the empirical ratios $\mu_{i,2k,k}$ to be independent for different points, one obtains an estimate of the intrinsic dimension $\hat{d}$ by numerically maximizing the mentioned likelihood.

For each (model, corpus, layer) combination, we perform an explicit scale analysis. To do so, we first estimate the ID while varying $k$. Then, by visual inspection, we select a suitable $k$ that coincides with a plateau in ID estimate (Denti et al., 2022). An example of such a scale analysis is reported in Appendix C, where we also demonstrate our estimates to be robust to changes in scale.

In order to delimit high-ID peaks across layers, we conventionally locate the end of the peak at the closest inflection point after its maximum value. The beginning of the peak corresponds, then, to the last layer before the maximum with value equal or greater than that at the end of the peak.

## 3.5 QUANTIFYING THE RELATIVE INFORMATION CONTENT OF DIFFERENT REPRESENTATIONS

We wish to relate dimensional expansion or compression of representations across layers to changes in their neighborhood structure. If neighborhood structure defines a semantics in representational space (Boleda, 2020), then layers whose activations have similar neighborhood structures perform similar functions.

In particular, the layers of an LM are iterative reconfigurations of representation space. We quantify the extent of reconfiguration (conversely, stability) using a statistical measure called Information Imbalance (Glielmo et al., 2022), hereon referred to as $\Delta$. Given two different spaces $A$ and $B$, this quantity, defined in equation 1, measures the extent to which the neighborhood ranks in space $A$ are informative about the ranks in space $B$ (since ID computation is based on Euclidean distance, ranks are also obtained with Euclidean distance):

$$\Delta(A \to B) = \frac{2}{N^2} \sum_{i,j|r_{ij}^A=1} r_{ij}^B. \tag{1}$$

In words, $\Delta$ is the average rank of point $j$ with respect to point $i$ in space $B$, given that $j$ is the first neighbour of $i$ in space $A$. If $\Delta(A \to B) \sim 0$, space $A$ captures full neighborhood information about space $B$. Conversely, if $\Delta(A \to B) \sim 1$, space $A$ has no predictive power on $B$.

---

[3]E.g., Ansuini et al. (2019); Tulchinskii et al. (2023); Valeriani et al. (2023).

As $\Delta$ is a rank-based measure, it can be used to compare spaces of different dimensionalities and/or distance measures. In our specific case, this property implies that $\Delta$ is robust to possible dimension misalignments between layers.

In comparing the layers' representation spaces, we also considered three alternative measures broadly used in deep net representation analysis (see Sucholutsky et al., 2024; Williams, 2024, for recent surveys): Doimo et al. (2020)'s neighborhood overlap measure, also used by Valeriani et al. (2023), Representational Similarity Analysis (Kriegeskorte et al., 2008), and linear CKA (Kornblith et al., 2019). Only the last measure resulted in interpretable results, broadly coherent with those obtained with $\Delta$ (Appendix H).

We choose to focus on $\Delta$ as the most principled measure for the investigation of LM layers, where we can make very little assumptions about the shape of the manifold. Moreover, crucially, $\Delta$, unlike the other measures we considered, is non-commutative with respect to its arguments. That is, it is asymmetric upon a swap of spaces: $\Delta(A \rightarrow B) \neq \Delta(B \rightarrow A)$. This feature allows us to capture *directional* information containment. For example, we will see below that $\Delta$ detects the asymmetric relation between Pythia/OPT and Lllama representations (Figure 4), and it allows us to study the degree to which information in a certain layer directionally predicts information in the first, next, or last layer (Figure 2 and Figure 3).

## 4  RESULTS

We find, in line with previous work, that LMs represent language on a manifold of low intrinsic dimension. Furthermore, the representational ID profile over layers reveals a characteristic phase of both geometric and functional significance, marking, respectively, a peak in ID and transition in between-layer neighborhood similarity ($\Delta$), and a transition to abstract linguistic processing.

### 4.1  EMERGENCE OF A CENTRAL HIGH-DIMENSIONALITY PHASE

Figure 1 (left) reports the evolution of the ID for all models, averaged across corpora partitions (for per-corpus results, see Appendix C). In line with previous work (Cai et al., 2021; Cheng et al., 2023; Tulchinskii et al., 2023; Valeriani et al., 2023), we first observe that the ID for all models is $\mathcal{O}(10)$, which lies orders of magnitude lower than the models' hidden dimension at $4096 \sim \mathcal{O}(10^3)$.

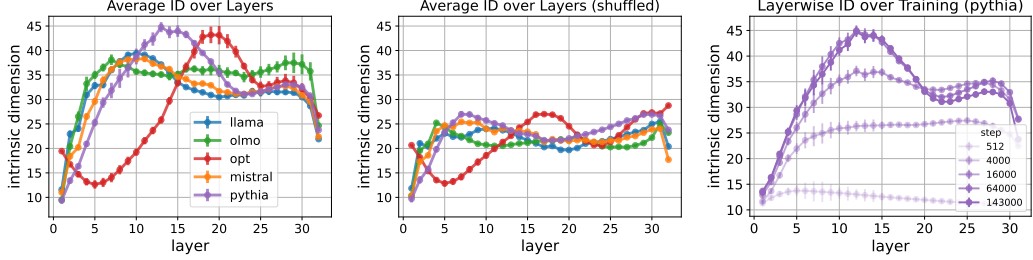

Figure 1: **Average ID over layers** over 5 random partitions from each of three corpora: Bookcorpus, the Pile and Wikitext. **(Left):** different models' layerwise ID plotted for the original corpora. **(Center):** different models' layerwise ID plotted for the shuffled corpora. **(Right):** different Pythia training checkpoints' layerwise ID on the original corpora, where darker curves are later checkpoints. In the middle layers, shuffled corpus ID (center) is *lower* than non-shuffled ID (left), suggesting that linguistic processing contributes to ID expansion. ID increases over the course of training for all layers to reach the final profile at step 143000 (right), suggesting that ID reflects learned linguistic features. All curves are shown with $\pm 2$ standard deviations (shuffled SDs are very small).

All models clearly go through a phase of high intrinsic dimensionality that tends to take place relatively early (starting approximately at layer 6 or 7, and mostly being over by layer 20), except for OPT, where it approximately lasts from layer 17 to layer 23. For all models, we also observe a second, less prominent peak occurring towards the end: only for OLMo, this second peak, which we do not analyze further, is as high as the first one. Appendix D reports ID profiles for a smaller and a

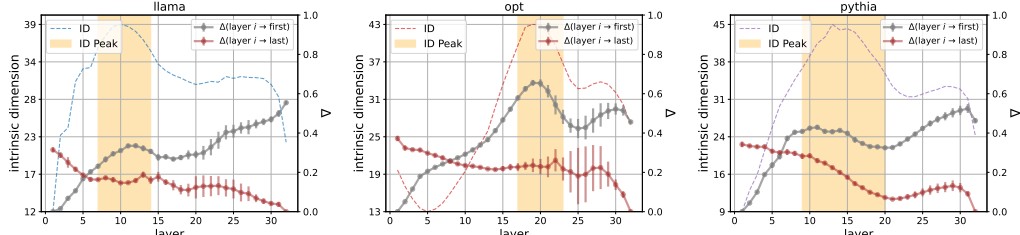

Figure 2: For Llama, OPT, Pythia (left to right), the ID is overlaid with $\Delta(l_i \to l_{first})$ (gray) and $\Delta(l_i \to l_{last})$ (brown). Plots are shown with $\pm 2$ standard deviations over 5 partitions of 3 corpora. For all models, there is a peak in $\Delta(l_i \to l_{first})$ (gray) around the ID peak.

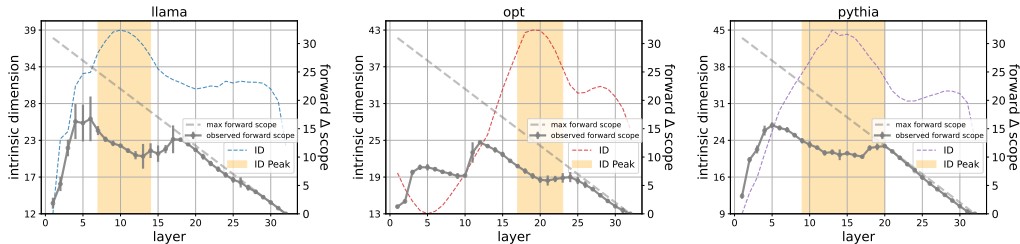

Figure 3: Forward $\Delta$ scope (left: Llama; center: OPT; right: Pythia): continuous lines report, for each layer $l_n$, the number of adjacent following layers $l_{n+k}$ for which $\Delta(l_n \to l_{n+k}) \leq 0.1$. The dashed line represents the longest possible scope for each layer. Values are averaged across corpora and partitions, with error bars of $\pm 2$ standard deviations.

larger LM from the Pythia family, showing that the presence of the ID peak is not dependent on the specific size range we are focusing on.

**ID is a geometric signature of learned structure** Figure 1 (center) shows that, when the models are fed shuffled corpora, ID remains low all throughout the layers. This suggests that the presence of high-ID peaks depends on the network performing meaningful linguistic processing. We further confirm this notion in Figure 1 (right), where we analyze the evolution of the ID profile over the course of training for Pythia (whose intermediate checkpoints are public). We find that, over the course of training, the IDs' magnitude not only grows over time, but that they become more peaked, indicating that the characteristic profile emerges from learned structure.

**The ID peak marks a transition in layer function** Figure 2 reports $\Delta(l_i \to l_{first})$ and $\Delta(l_i \to l_{last})$ for Llama, OPT and Pythia (with the remaining models in Appendix E, that also presents further analysis). We observe that the ID peak largely overlaps with a peak in $\Delta(l_i \to l_{first})$. Here, a large value of $\Delta$ implies that sequences which are nearest neighbours at the ID-peak are distant from each other in the input layer. That is, the ID peak layers no longer contain the information encoded in the initial representation of the sequence, suggesting they instead capture higher-level information. Meanwhile, we note that the $\Delta(l_i \to l_{last})$ profiles are not clearly related to the first ID peak, and we leave their analysis to future work.

Figure 3 shows the *forward $\Delta$ scope* profile for Llama, OPT and Pythia (see Appendix F for Mistral and OLMo). In particular, the plots indicate, at each source layer $l_n$, for how many contiguous following layers $l_{n+k}$ the quantity $\Delta(l_n \to l_{n+k})$ is below a low threshold (0.1). Qualitatively, this means that the source layer $l_n$ contains most of the information in the following $k$ layers. Coinciding with the ID peak is a downward dip in forward-scope, compatible with the interpretation that, while high-ID layers process similar information, this information differs from that of later layers.

**At the ID peak, different models share representation spaces** The high-dimensionality peaks contain similar information across models, as shown in the representative comparisons of Figure 4, which display cross-model $\Delta$ averaged across corpora and partitions (see Appendix G for the

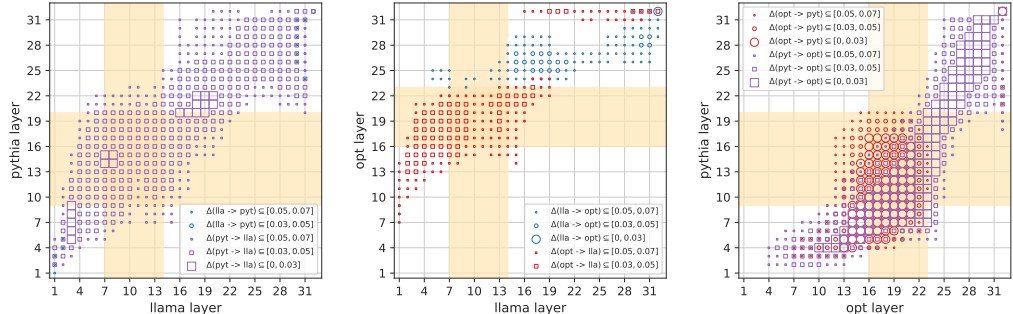

Figure 4: Cross-model $\Delta$. ID-peak sections are shaded in orange. Different symbols mark different $\Delta$ levels in the two directions (lower values correspond to a stronger trend towards information containment). High $\Delta$ scores ($> 0.1$), corresponding to low information containment, are not shown. Values averaged over corpora and partitions.

other pairs, and Appendix H for broadly comparable patterns emerging from CKA). The high-ID layer intersections always correspond to regions with a concentration of low-$\Delta$ values, indicating that, between models, representations at the peak have close neighborhood structures, and thus capture similar semantics (note, still, that low-$\Delta$ values can also occur outside the peak intersections, suggesting high ID might be a sufficient, rather than necessary, condition for cross-model similarity).

We also notice a marked asymmetry in which the ID-peak layers of Pythia and OPT, the two models with the highest absolute IDs, directionally contain other models' representations. On the other hand, when models have similar maximum IDs (such as in the Pythia vs. OPT comparison), $\Delta$ is more symmetric and very small, implying that their representation spaces are really equivalent.

## 4.2 Language processing during the high-dimensionality phase

We just saw, via geometric evidence ($\Delta$), that the high-ID phase marks a change in processing function. Now, we attempt to interpret *what* that function is. To do so, we look at layer-wise classification accuracies for the probing tasks described in Appendix I. We find that, in general, ID-peak representations fail at surface-form tasks but excel at semantic and syntactic tasks, indicating a functional transition from superficial to abstract linguistic processing.

**The ID-peak representations contain less surface-form information**  In Figure 5a, we consider the accuracy for two tasks, Sentence Length and Word Content, which test whether a layer retains information about superficial properties of the input. We observe that the ability to correctly reconstruct the length, measured in tokens, of the input sentence gets lost as we climb the network layers, and performance starts a sharp decrease at the onset of the ID peak or shortly before it. Intriguingly, for OPT (Figure 5a, center), which is the model with the latest ID peak, initial accuracy is relatively stable, and only starts to significantly decrease at the onset of the peak, further suggesting that it is only during the high-ID phase, even when the latter occurs relatively late, that superficial information (such as the number of tokens in the input sentence) is discarded by the models, that, as we are about to see, start instead at that point to process more abstract syntactic and semantic information.

Concerning the Word Content task, which tests the ability to detect the presence of specific words in the input, we generally observe a great decrease after the first few layers, particularly clear during ID expansion. Interestingly, accuracy tends to go up again after the ID peak, probably because, as the model prepares to predict the output, more concrete lexical information is again encoded in its representations. Together, the surface-form tasks confirm the evidence from $\Delta(l_i \rightarrow l_{first})$ (Figure 2 above) that the ID peak processes a more abstract type of information that, as we are about to see, may relate to the syntactic and semantic contents of the sequence.

**The ID peak marks a transition to syntactic and semantic processing**  Figure 5b reports accuracy for Llama, Pythia, and OPT (results for the remaining models in Appendix I) on the syntactic and semantic probe tasks. As described in detail in Appendix I, these tasks require a model to

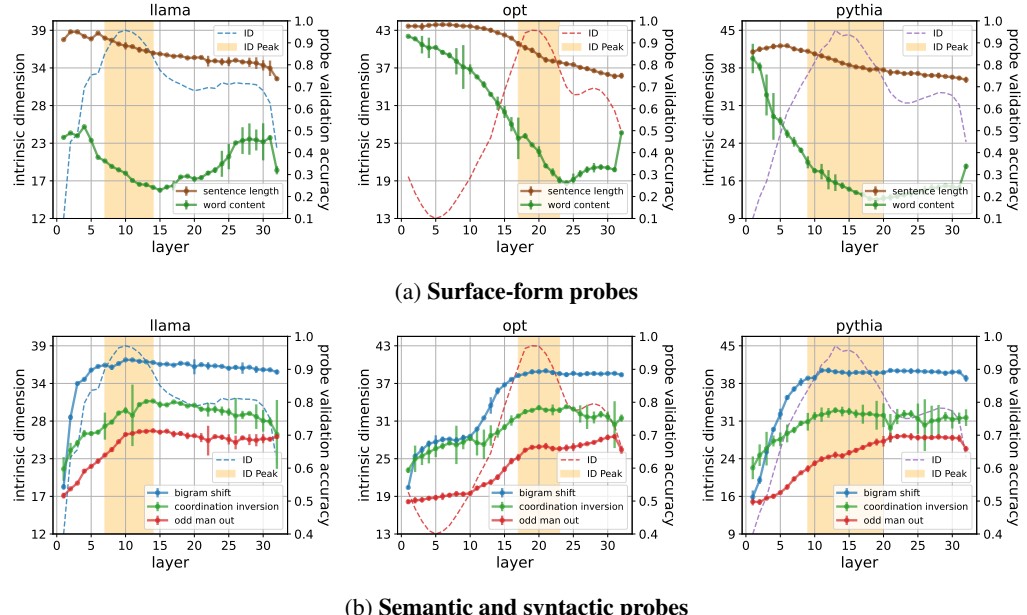

(a) **Surface-form probes**

(b) **Semantic and syntactic probes**

Figure 5: Linguistic knowledge probing performance ± 2 SDs across 5 random seeds is shown with the ID profile for Llama, OPT, and Pythia (left to right). Row (a) corresponds to surface-form tasks Sentence Length and Word Content, where probe performance decreases through the ID peak. Row (b) corresponds to syntactic and semantic tasks Bigram Shift, Coordination Inversion and Odd Man Out, where probe performance for all tasks attains maximum (or close) within the ID peak. This suggests the ID peak marks *abstract*, and not surface, representations of the input.

detect whether a sentence is syntactically well-formed (Bigram Shift), whether it is semantically coherent (Odd Man Out) and whether it describes a pair of events or states in a plausible order (Coordination Inversion). Despite variation across tasks and models, we observe that, in general, asymptotic accuracy is reached during the ID peak phase. Again, this implies that, for OPT, the asymptote is reached later. For OLMo (Appendix I), we observe in some cases a late accuracy peak, related to the second ID expansion phase that this model undergoes. In general, however, once top accuracy is reached, performance stays quite constant across layers, suggesting that the expressivity of high dimensionality permits rich linguistic representation of the inputs, but, once this information is extracted, it is propagated across subsequent layers. This is intuitive, as high-level linguistic information is useful to the network for its ultimate task of next-token prediction. Appendix D confirms the same patterns for a smaller and a larger Pythia model.

**Better LMs have higher ID peaks, earlier**    Given the apparent linguistic importance of the high-ID phase, a natural question concerns the extent to which the nature of the ID peak relates to the LM's performance on its original task of next-token prediction. The question was already partially answered by Figure 1, which shows an absence of peaks, respectively, when trained LMs process shuffled text and when untrained LMs process normal text; in both cases, next-token prediction is impossible.

We further computed Spearman correlations between average prediction surprisal for each (model, corpus) combination and the corresponding *maximum ID values* (Figure 6, left), as well as *ID-peak onsets* (Figure 6, right).[4]

As the plots show, even when limiting the analysis to sane text processed by fully trained models, where the differences in surprisal will be smaller, there is a marginal tendency for maximum ID value to inversely correlate with surprisal: the *higher* the peak, the *better* the LM is at predicting the next

---

[4]Due to its range, it's easier to visualize surprisal, instead of the more commonly used perplexity measure, in the plots. As perplexity is exponentiated surprisal, and we are using Spearman correlation (which is robust to monotonic transformations such as exponentiation), our results are not affected by this choice.

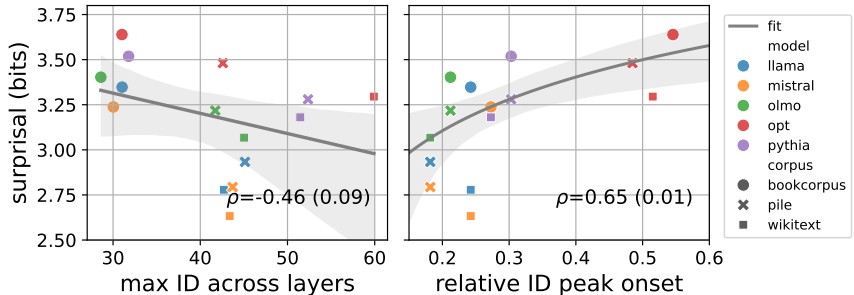

Figure 6: Surprisal plotted against the maximum ID across layers (left) and the relative ID peak onset over layers (right), where each datapoint is a (model, corpus) combination ($N = 50k$ sequences per corpus). A linear fit (left) and log-linear fit (right) are shown. **(Left):** Surprisal negatively correlates to maximum ID with Spearman $\rho = -0.46$, $p = 0.09$, meaning that *higher ID indicates better LM performance*. **(Right):** Surprisal positively correlates to ID peak onset, $\rho = 0.65$, $p = 0.01$, meaning that an *earlier ID peak indicates better LM performance*.

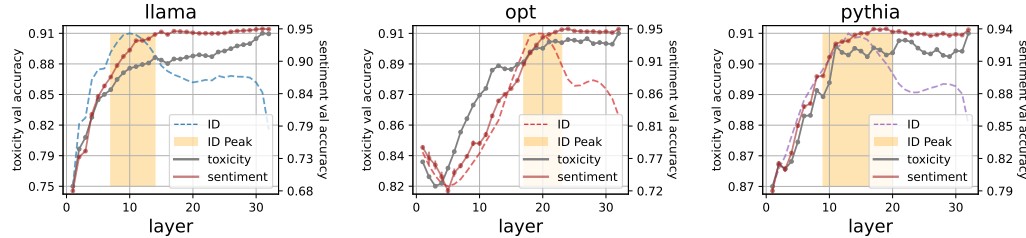

Figure 7: Validation performance of representation transfer performance on toxicity and sentiment classification for Llama, OPT, and Pythia (left to right). All validation accuracy curves are plotted with $\pm 2$ standard deviations over 5 random seeds. For all models, classification validation accuracy converges within the ID peak.

token. There is, moreover, a significant positive correlation between surprisal and the onset of the ID peak: the *earlier* the peak, the *better* the model is at predicting the next token.

We just saw that the ID peak might mark the phase where the model first completes a full syntactic and semantic analysis of the input. The earlier this analysis takes place, the more layers the model will have to further refine its prediction by relying on it. The correlation between ID peak onset and surprisal thus indirectly confirms the importance of the high-dimensional processing phase for good model performance.

**ID-peak layers are the first to transfer to downstream tasks**   Finally, given that the ID peak seems to mark the onset of more abstract linguistic processing of the input, representations at this peak should also viably transfer to downstream linguistic tasks. We confirm this hypothesis for sentiment and toxicity classification.

Validation accuracy curves for both tasks are shown for Llama, Pythia and OPT in Figure 7, with more results in Appendix J. Similar to the semantic and syntactic probing results, downstream classification performance consistently converges at the ID peak across models and remains high thereafter. Note that, while the ID peak is computed from three generic input corpora (Bookcorpus, Pile and Wikitext), it still predicts transferability to different downstream datasets that are reasonably in-distribution.

## 5   CONCLUSION

It is evident that a LM needs to extract information from its input to predict the next token. The non-trivial fact we show here is that this process does not gradually refine representations, but rather

undergoes a phase transition characterized by ID expansion, cross-model information sharing, and a switch to abstract information processing. Our main take-home message is that different LMs consistently develop a central-layer phase where the intrinsic dimension is expanded, which is the locus of deeper linguistic processing.

Our paper focused on detecting broad qualitative patterns. We found very consistent converging evidence from a variety of models, corpora and experiments (even OLMo, that is to some extent the "outlier" model, displays the ID peak and the relevant associated properties). Future work should establish a clearer causal connection between intrinsic dimension and language processing, via layer ablations studies and by constraining the dimensionality of different layers during language processing.

Mirroring Jastrzebski et al. (2018)'s proposition for visual networks, our findings are compatible with a view of LMs in which the high-dimensional early-to-mid layers functionally specialize to analyze inputs in a relatively fixed manner, whereas later layers may more flexibly refine the output prediction, using information extracted during the high-ID phase. Interestingly, two recent papers (Gromov et al., 2024; Men et al., 2024) found that (1) late LM layers better approximate each other than earlier layers do, and (2) pruning late layers (excluding the last) affects performance less than pruning earlier layers. This fully aligns with our results, and suggests a need to test the effect of pruning inside and outside the ID peak. Other studies have highlighted the importance of central layers in performing various core functions. For example, Hendel et al. (2023) find that compositional "task vectors" are formed in layers superficially consistent with the peaks we detect. Again, future work should more thoroughly study the relation between the ID profile and specific circuits detected in mechanistic interpretability work.

While our work closely relates to that of Valeriani et al. (2023), who studied the evolution of ID in vision and protein transformers, an interesting contrast is that they found crucial semantic information to coalesce during a dimensionality reduction phase, whereas we associated similar marks to a dimensionality expansion phase. This difference in observations may be partly attributed to our differing methodologies, since Valeriani and colleagues analyzed the ID of average sequence token and probed semantic information using a classifier based on nearest neighbors. In line with our results, recent work in theoretical neuroscience shows that high ID, thanks to its expressivity, underlies successful few-shot learning (Sorscher et al., 2022) and generalizable latent representations for DNNs (Elmoznino & Bonner, 2024; Wakhloo et al., 2024). Conversely, primarily in artificial and biological vision, low ID is linked to generalization thanks to representations' robustness to noise and greater linear separability in embedding space (Amsaleg et al., 2017; Chung et al., 2018; Cohen et al., 2020). Clearly, whether dimensionality is a curse or blessing to performance depends on the context of the learning problem. Reconciling, then, why and when high performance arises from reduced or expanded dimensionality remains an important direction for future work.

From a more applied perspective, we see various ways in which our discovery of a consistent high-ID phase could be useful. ID profiles emerge as significant blueprints of model behaviour that could be used as proxies of model quality. ID information can be used for model pruning, or to choose which layers to fine-tune, or for model stitching and other model-interfacing operations, such as training LM-based encoding models of the brain (Antonello & Cheng, 2024). These are all exciting directions for future work.

## 6 REPRODUCIBILITY STATEMENT

Please find corpora, code and a readme document to reproduce our experiments at https://github.com/chengemily1/id-llm-abstraction . We report our compute usage in Appendix A, and describe all publicly available resources we used in Appendix B. Appendix C, Appendix I and Appendix J specify relevant hyperparameter choices for ID computation, probing task and transfer task implementation, respectively.

## 7 ACKNOWLEDGMENTS

We thank the members of the COLT group, Mor Geva, the ICLR reviewers, and the area chair for useful feedback. This project has received funding from the European Research Council (ERC)

under the European Union's Horizon 2020 research and innovation program (grant agreement No. 101019291). This paper reflects the authors' view only, and the funding agency is not responsible for any use that may be made of the information it contains. Additionally, D.D. received support from the project "Supporto alla diagnosi di malattie rare tramite l'intelligenza artificiale"(CUP: F53C22001770002).

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

## A COMPUTING RESOURCES

All experiments were run on a cluster with 12 nodes with 5 NVIDIA A30 GPUs and 48 CPUs each.

Extracting LM representations took a few wall-clock hours per model-dataset computation. ID computation took approximately 0.5 hours per model-dataset computation. Information imbalance

computation took about 2 hours per model-dataset computation. Probing/transfer classifiers took up to 2 days per task.

Taking parallelization into account, we estimate the overall wall-clock time taken by all experiments, including failed runs, preliminary experiments, etc., to be of about 20 days.

## B  ASSETS

**Bookcorpus**  https://huggingface.co/datasets/bookcorpus ; license: unknown

**Pile-10k**  https://huggingface.co/datasets/NeelNanda/pile-10k ; license: bigscience-bloom-rail-1.0

**Wikitext**  https://huggingface.co/datasets/wikitext ; license: Creative Commons Attribution Share Alike 3.0

**Llama**  https://huggingface.co/meta-llama/Meta-Llama-3-8B ; license: llama3

**Mistral**  https://huggingface.co/mistralai/Mistral-7B-v0.1 ; license: apache-2.0

**OLMo**  https://huggingface.co/allenai/OLMo-7B ; license: apache-2.0

**OPT**  https://huggingface.co/facebook/OPT-6.7b ; license: OPT-175B license

**Pythia**  https://huggingface.co/EleutherAI/pythia-6.9b-deduped ; license: apache-2.0

**DadaPy**  https://github.com/sissa-data-science/DADApy ; license: apache-2.0

**scikit-learn**  https://scikit-learn.org/ ; license: bsd

**PyTorch**  https://scikit-learn.org/ ; license: bsd

**Probing tasks**  https://github.com/facebookresearch/SentEval/tree/main/data/probing ; license: bsd

**Toxicity dataset**  https://huggingface.co/datasets/google/jigsaw_toxicity_pred ; license: CC0

**Sentiment dataset**  https://huggingface.co/datasets/stanfordnlp/imdb ; license: unknown

## C  INTRINSIC DIMENSION

### C.1  SCALE ANALYSIS

In this section we explicitly show how to perform a scale analysis in order to select a proper and meaningful scale when computing the ID. As explained in the main text, ID estimation might be affected by undesirable effects that are always present when dealing with real-world datasets and that can hide the true dimensionality of the manifold underlying the data. Such effects include the presence of noise, which typically affects small scales, and density variations and manifold curvature, which, instead, tend to be observed at a larger scale (Denti et al., 2022). It is thus recommended to choose an intermediate scale, where the ID estimate might be stable across different neighbourhood sizes, and it is less likely to be affected by the aforementioned spurious factors. For this reason, it is necessary to see how the ID changes when varying the neighbourhood size taken into account when performing the ID computation. We rely on the GRIDE estimator, which allows to explicitly select the number of neighbours considered. In particular, the rank of the first nearest neighbour used to compute the distance ratio is a hyperparameter, and we refer to the chosen value for this hyperparameter as the *scale* in what follows.

We performed scale analysis for each model and corpus. An example is provided in Figure C.1, which shows the results for Pythia on Pile. We plot the ID estimate for increasing scales (i.e., number of neighbours, x-axis in Figure C.1) for some representative layers. The rank of the neighbour considered is explored by means of powers of 2 in order to look at regions where the variation in ID is noticeable. The true ID is likely to lie at a scale in which the ID estimate approximately plateaus (Denti et al., 2022), which is marked by the highlighted region. For simplicity, per model-corpus combination, we choose one scale for all layers: in this particular example, we choose $k = 2^5$. In general, the optimal neighbourhood size tends to be around $k = 32$ for all (model, corpus) combinations, allowing us to reliably compare them (see Table C.1 for all $k$). Once the scale is chosen for each (model, corpus) combination, we plot the scale-adjusted ID estimates (see, e.g., Figure C.3).

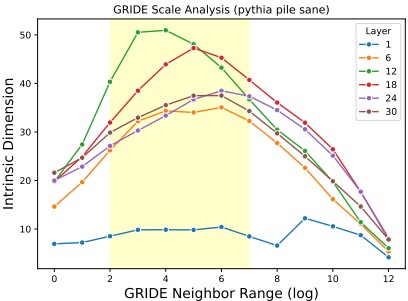

Figure C.1: GRIDE ID estimation neighbourhood scale analysis example for Pythia on the Pile on a single random seed, where each line is a layer's ID estimate at different scales. All layers shown reach a plateau in the highlighted range.

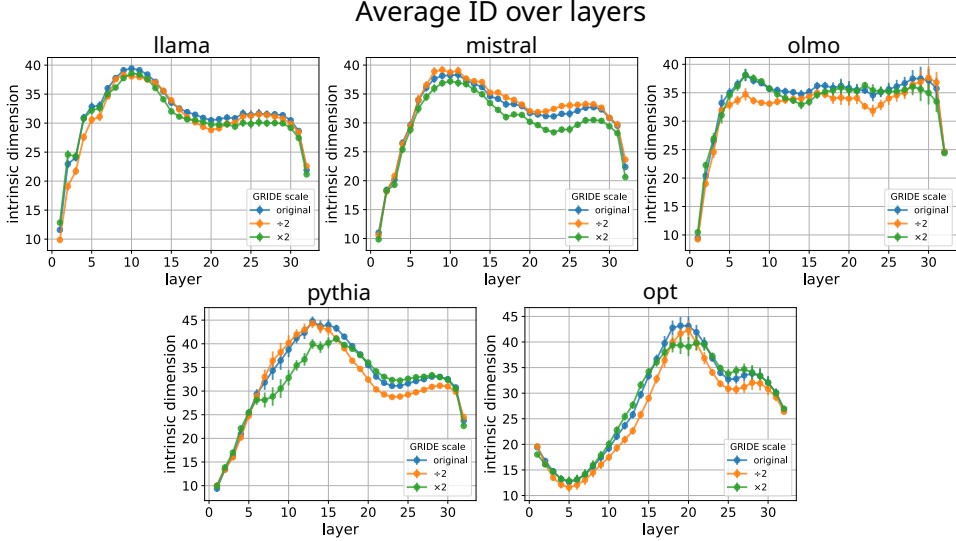

Figure C.2: Robustness of the scale analysis within the plateau region across models. The ID estimates are only weakly affected by doubling (green) or halving (orange) the scale with respect to the results reported in the main text (blue).

As a further robustness test, we show in Figure C.2 that results are fairly stable to choosing scales within the shaded range. For each model, we include a panel with the original chosen scale (in blue), e.g., $k = 2^5 = $32th nearest neighbor, and then look at the shape of the ID if we move the scale one (logarithmic) step down $k = 2^4 = $16 (orange) or up $k = 2^6 = $64 (green). Fortunately, we can see that the shapes of the resulting ID curves are stable across all models, and the ID values themselves change very little.

## C.2   ADDITIONAL RESULTS

Figure C.3 displays the evolution of estimated ID, scale-adjusted, per layer for all corpora and models. While the magnitude of ID differs across corpora, with ID on Bookcorpus lower than those of the Pile and Wikitext for all models, all corpora exhibit the characteristic high-ID peak, and at nearly the same onset. The dampened Bookcorpus peak IDs, which are still significantly above the corresponding shuffled ID peaks, might be explained by the fact that this corpus is entirely made of novels (and lower-cased), and it is thus the less in-domain of the corpora we explored.

| model | corpus | mode | GRIDE $k$ |
|---|---|---|---|
| llama | bookcorpus | sane | 32 |
| | | shuffled | 128 |
| | pile | sane | 64 |
| | | shuffled | 128 |
| | wikitext | sane | 64 |
| | | shuffled | 16 |
| mistral | bookcorpus | sane | 64 |
| | | shuffled | 128 |
| | pile | sane | 128 |
| | | shuffled | 256 |
| | wikitext | sane | 64 |
| | | shuffled | 32 |
| olmo | bookcorpus | sane | 32 |
| | | shuffled | 8 |
| | pile | sane | 32 |
| | | shuffled | 128 |
| | wikitext | sane | 32 |
| | | shuffled | 16 |
| opt | bookcorpus | sane | 16 |
| | | shuffled | 16 |
| | pile | sane | 32 |
| | | shuffled | 16 |
| | wikitext | sane | 16 |
| | | shuffled | 16 |
| pythia | bookcorpus | sane | 32 |
| | | shuffled | 32 |
| | pile | sane | 32 |
| | | shuffled | 64 |
| | wikitext | sane | 32 |
| | | shuffled | 16 |

(a) GRIDE scale $k$ reported for each model, corpus, mode (in sane and shuffled) combination. For simplicity, we chose one $k$ for all layers.

| corpus | Pythia step | GRIDE $k$ |
|---|---|---|
| bookcorpus | 512 | 32 |
| | 4000 | 32 |
| | 16000 | 32 |
| | 64000 | 32 |
| pile | 512 | 4 |
| | 4000 | 32 |
| | 16000 | 32 |
| | 64000 | 64 |
| wikitext | 512 | 4 |
| | 4000 | 32 |
| | 16000 | 32 |
| | 64000 | 32 |

(b) GRIDE $k$ for additional Pythia checkpoints.

| corpus | Pythia size | GRIDE $k$ |
|---|---|---|
| wikitext | 2.8b | 32 |
| | 12b | 16 |

(c) GRIDE $k$ for additional Pythia sizes, on Wikitext dataset.

Table C.1: Gride IDs for each model, dataset, and mode

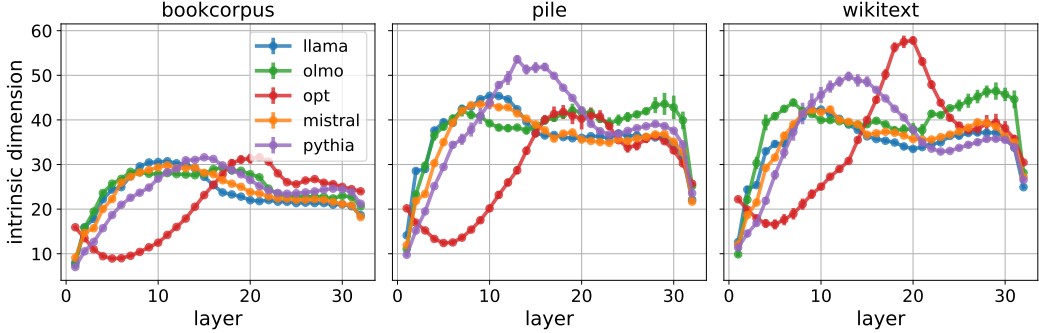

Figure C.3: ID evolution over layers, shown with one standard deviation (over corpus partitions), for all models and corpora (left to right: Bookcorpus, the Pile, and Wikitext).

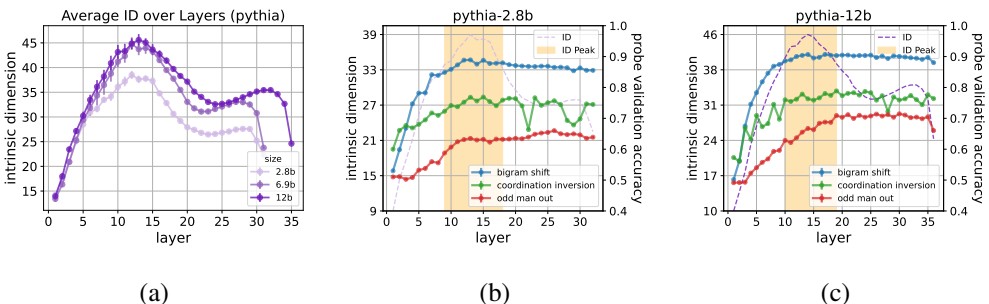

        (a)                           (b)                         (c)

Figure D.1: ID profiles (a) and probing-task experiments (b, c) reproduced with a smaller and a larger model from the Pythia family.

## D   INFLUENCE OF MODEL SIZE ON ID AND PROBING TASKS

We reproduce key experiments from the paper with a smaller (2.8b) and a larger (12b) version of the Pythia's architecture, suggesting that our findings are not limited to the scale of the models we analyzed in the paper.

**ID profiles**   We have plotted the ID profiles of smaller (2.8b) and larger (12b) Pythia models in Figure D.1-(a). Analogously to the evolution of ID during training (cf. Figure 1 right), we observe a similar profile across model sizes, with the ID magnitude of the peak growing with model size. Note that the ambient dimensionalities of the 2.8b, 6.9b, and 12b models are respectively 2560, 4096, and 5120. While the relative change in peak magnitude is much more dramatic between 2.8b and 6.9b than between 6.9b and 12b, the change is nevertheless not proportional to the relative change in hidden dimension, echoing the observation by Cheng et al. (2023) that ID saturates as ambient dimensionality increases.

**Probing tasks**   We also reproduce the probing-task experiments with the same smaller (2.8b) and larger (12b) models from the Pythia family, showing that the semantic/syntactic probing-task asymptote is reached within the span of the ID peak also at these newly tested sizes (see Figure D.1-(b,c)).

## E   INFORMATION IMBALANCE WITH RESPECT TO FIRST/LAST LAYER

Figure E.1 shows averaged $\Delta(l_i \rightarrow l_{first})$ (gray) and $\Delta(l_i \rightarrow l_{last})$ (brown) for Mistral and OLMo. Recall that the closer $\Delta(A \rightarrow B)$ is to 0, the more predictive $A$'s local neighborhood structure is of $B$'s. As expected, $\Delta(l_i \rightarrow l_{first})$ generally increases with $i$ as we go deeper in the layers; the reverse is true for $\Delta(l_i \rightarrow l_{last})$. However, $\Delta(l_i \rightarrow l_{first})$ appears to locally peak and plateau

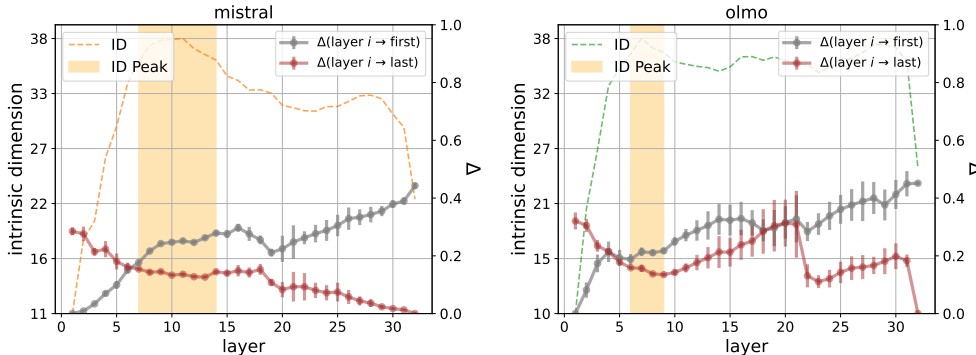

Figure E.1: For Mistral (left) and OLMo (right), the ID (hued) is overlaid with $\Delta(l_i \to l_{first})$ (gray) and $\Delta(l_i \to l_{last})$ (brown). Plots are shown with $\pm$ 2 standard deviations over corpora and partitions.

around the representational ID peak; that is, the ID expansion marks a phase of low predictivity from the intermediate layer to the input. OLMo's pattern is not as clear, and we might observe a second local information imbalance peak in proximity to the second ID peak characterizing this model.

Figure E.2 shows, in the spirit of the *Information Plane* (Shwartz-Ziv & Tishby, 2017), the $\Delta(l_i \to l_{last})$ plotted against $\Delta(l_{first} \to l_i)$ for all models across the layers. The trajectory reveals the dynamic evolution of input and output informativity along the residual stream. Consistently across models, ID peak layers (red) mark a change point in processing patterns. Pre-peak layers show a rapid increase along the x-axis but slow decrease along the y-axis, indicating a rapid departure from the inputs while not learning much about the outputs. This reflects the phase of *semantic abstraction*. Post-peak layers' information dynamics vary by model, though all are marked by a final rapid decrease in $\Delta(l_i \to l_{last})$ and a decrease in $\Delta(l_{first} \to l_i)$. This suggests a return to *surface-level* information processing in order to predict the next token.

## F  Forward $\Delta$ scope

Figure F.1 reports the forward $\Delta$ scope profile for the remaining two LMs not shown in the main text (Mistral and OLMo).

## G  Cross-model $\Delta$

Figure G.1 show cross-model $\Delta$ for the remaining combinations, confirming that there are areas of low cross-model information imbalance at the intersection of the high-ID peaks. In combinations involving OLMo, we observe a tendency for low $\Delta$ to stretch along the other LM high-ID section, suggesting that the high-ID layers of other LMs share information with a wider range of OLMo layers.

## H  Cross-model layer comparison using CKA

We reproduce the cross-model layer comparison experiments of Figure 4 using linear CKA (Kornblith et al., 2019) as an alternative measure of similarity. Figure H.1 broadly confirms the results found with $\Delta$.

## I  Probing tasks

### I.1  Tasks

We use the following classification tasks from Conneau et al. (2018):

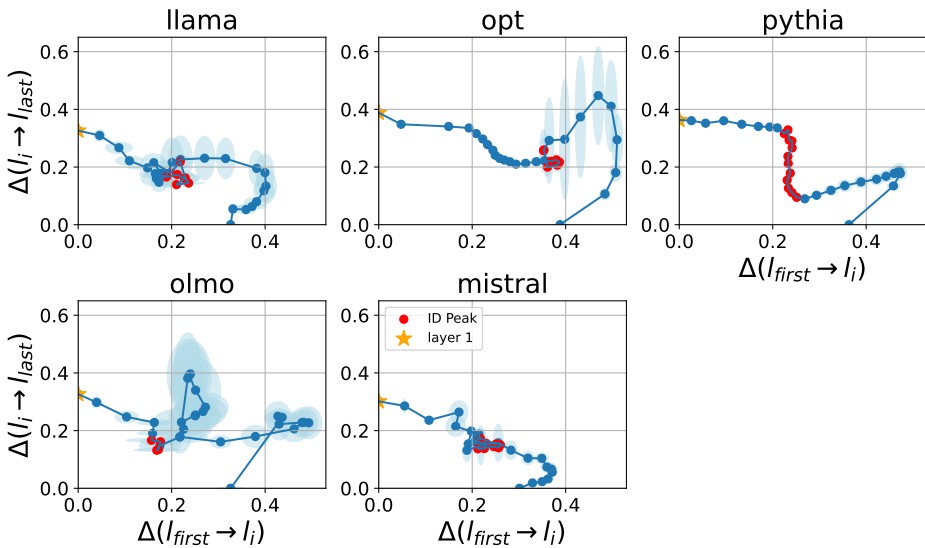

Figure E.2: $\Delta(l_i \to l_{last})$ **vs.** $\Delta(l_{first} \to l_i)$. For each LM, the information imbalance $\Delta$ from the $i$th to the last layer is plotted against $\Delta$ from the first to the $i$th layer, where **lower $\Delta$ means more informative**. Each point on the curve corresponds to a single layer $i$; for all models, the first layer (gold star) begins on the y-axis, $\Delta(l_{first} \to l_i) = 0$, and the trajectory through the layers ends on the x-axis, $\Delta(l_i \to l_{last}) = 0$. Consistently across models, ID peak layers (red) mark a changepoint in processing patterns. Pre-peak layers show a rapid increase along the x-axis but slow decrease along the y-axis, indicating a rapid departure from the inputs while not learning much about the outputs. This reflects the phase of *semantic abstraction*. Post-peak layers' information dynamics vary by model, though all are marked by a final rapid decrease in $\Delta(l_i \to l_{last})$ and a decrease in $\Delta(l_{first} \to l_i)$. This suggests a return to *surface-level* information processing in order to predict the next token.

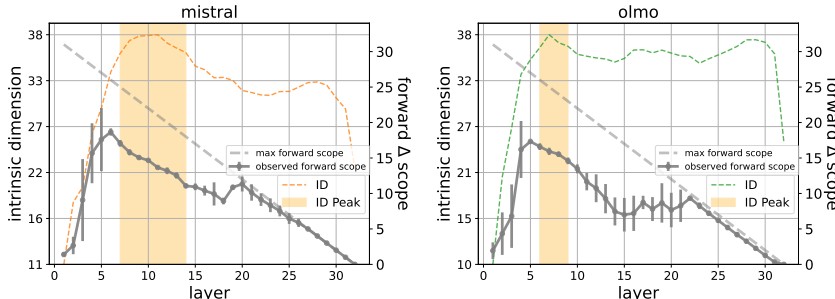

Figure F.1: Forward $\Delta$ scope (Mistral, Olmo): continuous lines report, for each layer $l_n$, the number of adjacent following layers $l_{n+k}$ for which $\Delta(l_n \to l_{n+k}) \le 0.1$. The dashed line represents the longest possible scope for each layer. Values are averaged across corpora and partitions, with error bars of $\pm 2$ standard deviations.

- *Surface form*

  **Sentence Length**  Predict input sentence length in tokens (lengths binned into 5 intervals).
  **Word Content**  Tell which of a pre-determined set of 1k words occurs in the input sentence.

- *Syntax*

  **Bigram Shift**  Tell whether the input sentence is well-formed, or it has been corrupted by inverting the order of two adjacent tokens (e.g., "*They were **in present** droves, going from table to table and offering to buy meals, drinks or generally attempting to strike up conversations*").

- *Semantics*

  **Coordination Inversion**  Tell whether a sentence is well-formed or it contains two coordinated clauses whose order has been inverted (e.g., "*Then I decided to treat her just as I would anyone else, but at first she'd frightened me*").
  **Odd Man Out**  Tell whether a sentence is well-formed, or it involves the replacement of a noun or verb with a random word with the same part of speech (e.g., "*The people needed a sense of **chalk** and tranquility*").

We exclude the following tasks because we observed ceiling effects across the layers, suggesting that the models could latch onto spurious correlations in the data: Past Present, Subject Number and Object Number. We exclude Top Constituents and Tree Depth because they produced hard-to-interpret results that we believe are due to the fact that they rely on automated syntactic parses that are not necessarily consistent with the way modern LMs process their inputs.

### I.2 SETUP

We use the training and test data provided by Conneau et al. (2018). We train a MLP classifier for each task and each layer of each LM, repeating the experiment with 5 different seeds.

We fixed the following hyperparameters of the MLP, attempting to approximate those used in the original paper (as each task takes days to complete, we could not perform our own hyperparameter search):

- Number of layers: 1
- Layer dimensionality: 200
- Non-linearity: logistic
- L2 regularization coefficient: 0.0001
- seeds: 1, 2, 3, 4, 5

For all other hyperparameters, we used the default values set by the the Scikit-learn library (Pedregosa et al., 2011).

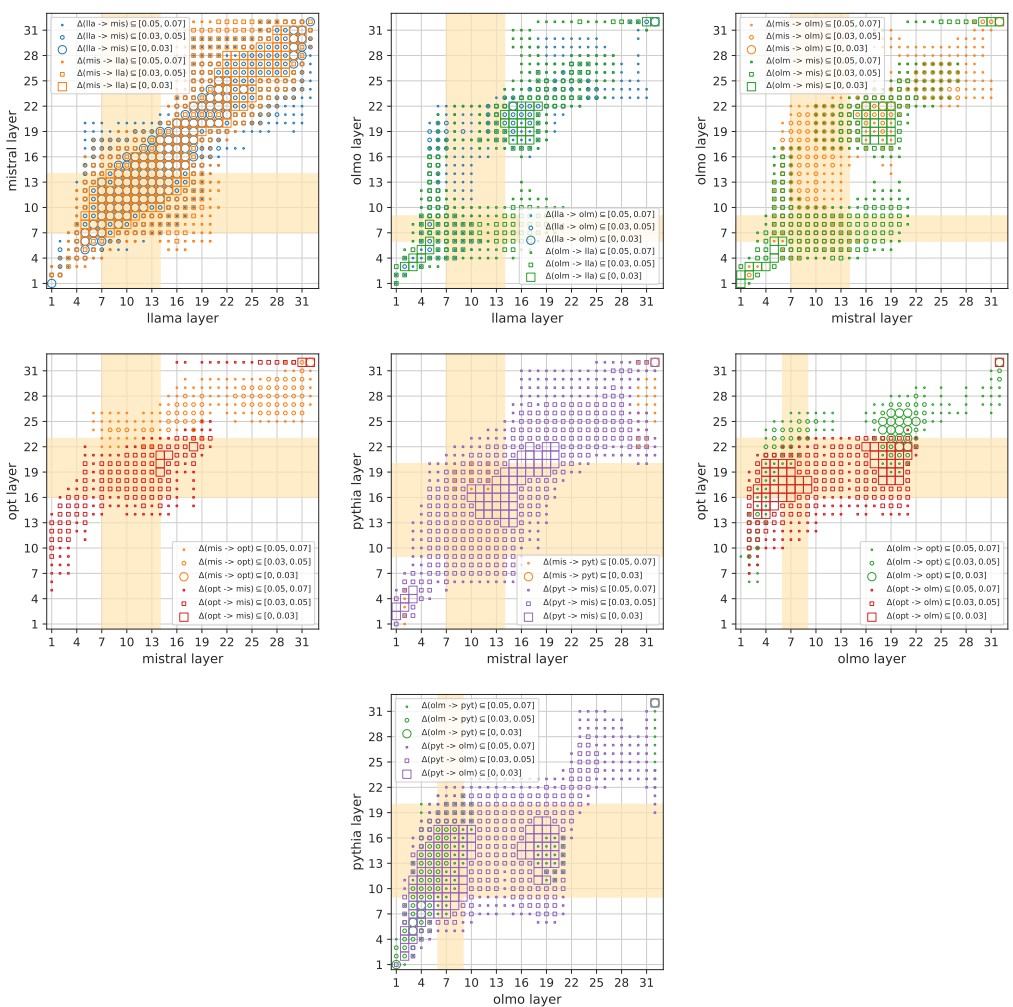

Figure G.1: Cross-model $\Delta$. ID-peak sections are shaded in orange. Different symbols mark different information imbalance levels in the two directions (the lower the $\Delta(A \rightarrow B)$ values, the more the information in $B$ is contained in $A$). High imbalances ($> 0.1$) are not shown. Values averaged across corpora and partitions.

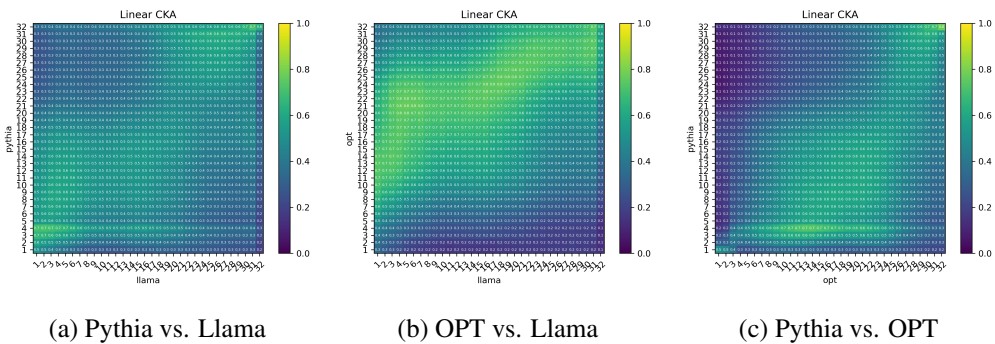

Figure H.1: Cross-model layer similarity measured using linear CKA. Like in Figure 4, ID-peak intersections tend to coincide with high-similarity areas.

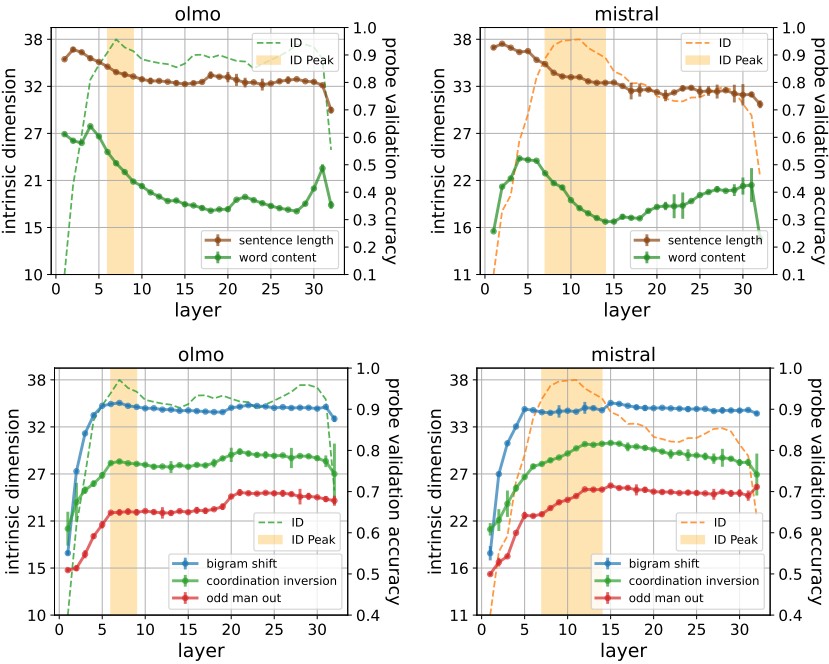

Figure I.1: Linguistic knowledge probing performance $\pm 2$ SDs across 5 random seeds is plotted with the ID across layers for OLMo and Mistral (left to right). (Top row) Surface form tasks Sentence Length and Word Content, where probe performance decreases through the ID peak. (Bottom row) Semantic and syntactic tasks Bigram Shift, Coordination Inversion and Odd Man Out, where probe performance for all tasks attains maximum within the ID peak.

We repeated all experiments after shuffling the example labels (both at training and test time). This provides a baseline for each task, as well as functioning as a sanity check that a probing classifier is not so powerful as to simply memorize arbitrary patterns in the representations (Hewitt & Liang, 2019). Note that, since, as shown in Figure I.2, performance is essentially constant (and at chance) on shuffled labels, accuracy is in our case equivalent to selectivity, the measure recommended by Hewitt & Liang (2019). We thus focus on accuracy, as the more familiar measure.

## I.3    ADDITIONAL RESULTS

Figure I.1 reports the probing performance for OLMo and Mistral, and for surface form (top row) as well as syntactic and semantic tasks (bottom row). As for Llama, OPT, and Pythia, we observe that the first ID peak converges to viable syntactic/semantic abstraction of inputs, while discarding information about surface form.

Figure I.2 shows that, on the shuffled corpora ablation, the MLP probes perform at chance. We then confirm that semantic and syntactic information is contained in the *model representations* and not in the probes.

## J    DOWNSTREAM TASKS

### J.1    TASKS

**Toxicity detection.**    We use a random balanced subset ($N = 30588$) of Kaggle's jigsaw toxic comment classification challenge (Adams et al., 2017), where each data point consists of a natural language comment and its binary toxicity label.

**Sentiment classification.**    We use a dataset of IMDb movie reviews (Maas et al., 2011), where each data point consists of a natural language film review and a corresponding label $\in$ {positive, negative}.

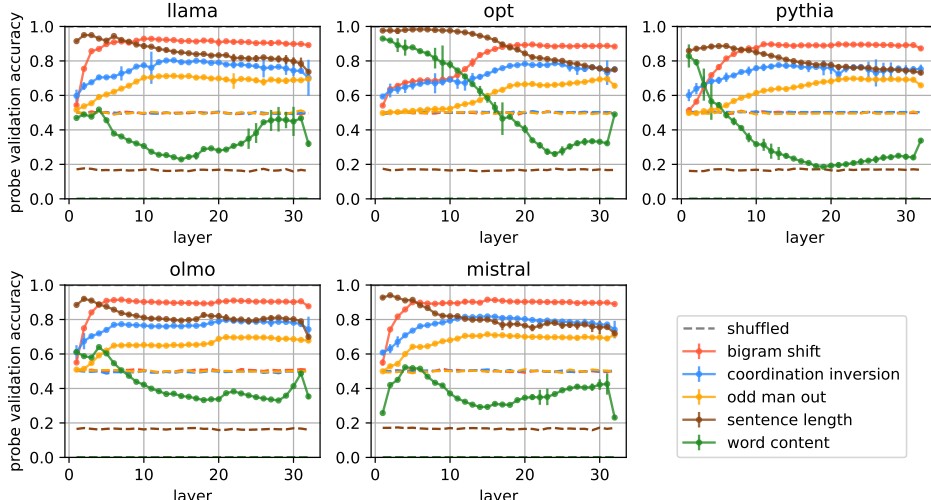

Figure I.2: Linguistic probe validation accuracy for semantic, syntactic, and surface tasks (solid lines) and their shuffled versions (dashed lines) are shown across layers for all models. The probing performance on shuffled corpora is constant at chance for all tasks and models.

To train the linear probes, we take a sample of size $N = 25000$ corresponding to the train split on HuggingFace. We repeat the experiment with 5 distinct seeds.

## J.2 SETUP

To train the linear probes, we first divide the data at random into train (80%) and validation (20%) sets. We feed each training set through each model and gather the last token hidden representations at each layer. Then, using PyTorch (Paszke et al., 2019), we train one linear probe per layer with hyperparameters as follows,

- Number of epochs: 1000
- lr: 0.0001
- seeds: 32, 36, 42, 46, 52

and we report the best validation accuracy over 5 random seeds.

## J.3 ADDITIONAL RESULTS

Similar to Llama, OPT, and Pythia, Figure J.1 shows that validation performance for downstream tasks for Mistral and OLMo converge in the ID peak.

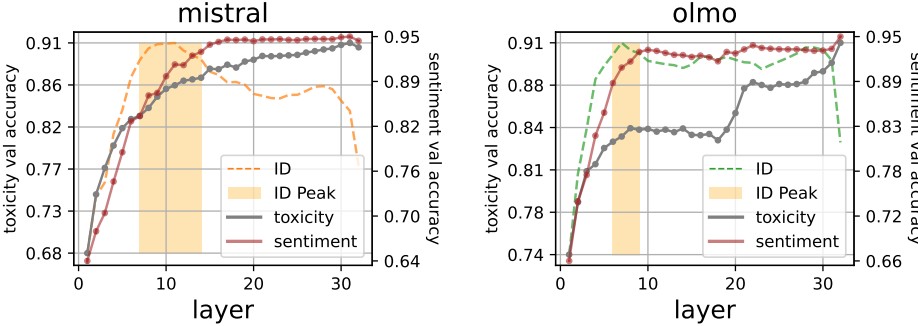

Figure J.1: Validation performance on downstream transfer tasks on toxicity and sentiment classification for Mistral (left) and OLMo (right). All validation accuracy curves are plotted with $\pm 2$ standard deviations over 5 random seeds. For all models, classification validation accuracy converges within the ID peak.

