# OpenReview forum: "Emergence of a High-Dimensional Abstraction Phase in Language Transformers"
_ICLR.cc/2025/Conference — ICLR 2025 Poster_

### Official Review · Reviewer_5snc · 2024-10-30

**Soundness:** 2
**Presentation:** 3
**Contribution:** 3
**Rating:** 5
**Confidence:** 3

**Summary:**

This work analyzes properties of representations of several LLMs through a few approaches: downstream probing, intrinsic dimensionality and information imbalance. The analysis is mainly developed around intrinsic dimension and they show LLMs typically have a few intrinsic dimension peaks across layers. Additionally, they suggest that those peaks indicate transition to abstract linguistic processing through a variety of analysis

**Strengths:**

I like that authors combine evidence from a few different perspectives to demonstrate the relation between intrinsic dimension peak and transition to abstract processing. They also conduct experiments on a few corpus and a few models as well, which make the claim more general and robust

**Weaknesses:**

The method section is weak and the explanation of intrinsic dimension computing is not enough given its importance in this work. I was not able to identify which variable is corresponding to the intrinsic dimension without going through the cited paper. It seems to be the variable $d$ which authors did not explain what it is.

Additionally, author made a wrong claim in line 177~178 that \mu has a generalised pareto distribution. I cannot find any resources claiming this specific distribution is a (generalized) pareto distribution including the original cited paper.

**Questions:**

In the data section, it is not very clear to me what does it mean to "extract 5 non-overlapping partitions of 10k 20-token sequences" and how the shuffled version is generated, can authors explain more about this?

In section: The ID peak marks a transition in layer function, I think the relation between ID peak and \delta(l_i \to l_first) is not very clear. It has very similar shape in OPT and somewhat in pythia, but LLAMA has a completely different curve. It is maximizing towards the end of the layer instead of the center of layer.

In section 4.2, authors also claim the relation between ID-peak and a few tasks. However, Figure (a) and (b) do not have very clear co-related trend between ID peaks and tasks' performance. In particular, task performance in Figure 5(b) seems to be monotonically increasing instead of peaking in the middle. Can authors justify more about this?

---

> ### Author Response · Authors · 2024-11-14
> **Responses to Weaknesses**
>
> Thanks for your careful review! Here are some clarifications that resolve the points raised in the Weaknesses. Please let us know if there are more specific things we can do to clarify the ID computation, such as restate the Theorems in Denti et al., 2022.

---

> > ### Author Response · Authors · 2024-11-14
> >
> > __I was not able to identify which variable is corresponding to the intrinsic dimension without going through the cited paper. It seems to be the variable $d$ which authors did not explain what it is__
> >
> > Thanks for pointing this out, that was our oversight! We have added a line in Section 3.4 that defines d to be the estimated ID.

---

> > > ### Author Response · Authors · 2024-11-14
> > >
> > > __wrong claim in line 177~178 that \mu has a generalised pareto distribution. I cannot find any resources claiming this specific distribution is a (generalized) pareto distribution including the original cited paper.__
> > >
> > > Theorem 2.2 in Denti et al, 2022 establishes that the $\mu_i$ follow a generalized Pareto distribution. In particular, please see Eq (9) in Denti et al, 2022 and their Supplementary Materials for the proof. If the reviewer suggests, we can restate Theorems 2.2 and 2.3 from Denti et al., 2022 in our paper to make that more clear.

---

> > > > ### Author Response · Authors · 2024-11-14
> > > >
> > > > __In the data section, it is not very clear to me what does it mean to "extract 5 non-overlapping partitions of 10k 20-token sequences" and how the shuffled version is generated, can authors explain more about this?__
> > > >
> > > > For each corpus, we sampled without replacement a total of 50k distinct 20-token sequences. We then divided these into partitions of 10k sequences each, which we used for the experiments. For the shuffled version, we first randomized the order of the tokens in each corpus, and then proceeded as with the non-randomized versions. We will clarify this in the paper.

---

> > > > > ### Author Response · Authors · 2024-11-14
> > > > >
> > > > > __In section: The ID peak marks a transition in layer function, I think the relation between ID peak and \delta(l_i \to l_first) is not very clear. It has very similar shape in OPT and somewhat in pythia, but LLAMA has a completely different curve. It is maximizing towards the end of the layer instead of the center of layer.__
> > > > >
> > > > > Are you referring here to Figure 2? In this figure, for all models, \delta(l_i \to l_first) reaches a (local) peak within the ID peak span, followed by a local minimum either at the very end of the ID  peak span or right after it. We would be grateful for further clarifications about the difference you see.

---

> > > > > > ### Author Response · Authors · 2024-11-14
> > > > > >
> > > > > > __In section 4.2, authors also claim the relation between ID-peak and a few tasks. However, Figure (a) and (b) do not have very clear co-related trend between ID peaks and tasks' performance. In particular, task performance in Figure 5(b) seems to be monotonically increasing instead of peaking in the middle. Can authors justify more about this?__
> > > > > >
> > > > > > We agree that the patterns we observe can be slightly noisy; still, _for all tasks and models_, we see in Figure 5(b) a phase of fast increase in task performance that ends under the ID peak. After this phase, task performance tends to plateau: it remains constant, or it increases or decreases very slowly. This is in accordance with our interpretation that the ID-peak-span is where the model performs a full linguistic processing of the input, resulting in representations that contain useful information for the syntactic/semantic probing tasks, as well as downstream tasks (sentiment/toxicity).
> > > > > >
> > > > > > Note that we do not claim that, once this linguistic processing is performed, the model gets rid of the information it provided: indeed, we expect it to use it in the subsequent layers to further refine its analysis and make its guess about the continuation. This is fully compatible with the patterns we observe: we would indeed be surprised if the probing-task performance dramatically decreased after the peak.

---

> > > > > > > ### Author Response · Authors · 2024-12-02
> > > > > > >
> > > > > > > Thank you again for taking the time to review our paper. As the discussion period ends today, we kindly ask that you confirm whether all your questions (especially concerning methodology) have been resolved. Please let us know if anything remains unclear!

---

### Official Review · Reviewer_61Jw · 2024-11-04

**Soundness:** 2
**Presentation:** 3
**Contribution:** 2
**Rating:** 6
**Confidence:** 4

**Summary:**

The submission is analytic work trying to correlate intrinsic dimension of intermediate NN/LLM representations with linguistic targets. Comparison of the method applied to different models allows insights into some of their learned structural differences.

**Strengths:**

The findings in bold letters at lines 406-407 and lines 425-426 may be useful to some researchers that need to train and/or select models. The fact that ID seems to change gradually over layers is interesting, but may have a simple explanation in the extreme averaging scale of these models.

**Weaknesses:**

Except for the few strengths mentioned above, the submission does not explain for what else the gained insights can be used for or wether they are more useful than that at all.

The analysis focuses only on fully trained models and does not provide insights into how ID changes over time. A correlation analysis to other work would add more value. My first thought was a correlation to the IB method (e.g. Tishby et al. 2000 and Schwartz-Ziv & Tishby 2017), but this may not be the only or best choice.

The submission wrongly mentions PCA being linear (line 061, applies only to its original form) which leads to the quick conclusion to discard it. This is puzzling as the research on non-linear PCA is quite diverse based on very different techniques and there's even early work using neural networks dating back to 1991 (Mark Kramer: "Nonlinear PCA Using Autoassociative NNs").

**Questions:**

I strongly advice to improve the submission w.r.t. the mentioned weaknesses. That helps both quality and reach.

The first paragraph of Asset Section C.1 (lines 809-836, in particular 828-829) mentions sensitivity of ID estimation w.r.t. to noise, small scales, density variations and curvature. That analysis suggests some sort of frequency decomposition integrated with the ID estimation.

---

> ### Author Response · Authors · 2024-11-15
> **Response to Weaknesses**
>
> Thanks so much for your thoughtful feedback! Please find below our responses to your comments in the Weaknesses:

---

> > ### Author Response · Authors · 2024-11-15
> >
> > __the submission does not explain for what else the gained insights can be used for or wether they are more useful than that at all.__
> >
> > Thanks for this point; we respectfully disagree. In the Conclusion lines 511-515, we detail a path forward using our insights: "ID profiles emerge as significant blueprints of model behaviour that could be used as proxies of model quality. ID information can be used for model pruning, or to choose which layers to fine-tune, or for model stitching and other model-interfacing operations."
> >
> >
> > We can elaborate on the latter. Our results linking the ID peak to feature richness, abstraction, and inter-LM representational similarity provides a guideline for model interfacing. For instance, this would recommend ID peak representations as the first viable ones for semantic/syntactic downstream tasks and model stitching. And, it recommends ID peak representations for building CLIP-like models. In addition, an exciting real example of our results’ usefulness is in building encoding models of the brain. In this domain, follow-up work to our paper has already found ID peak layers to best model fMRI responses to natural language. We will discuss these important points more extensively in the final version.

---

> ### Author Response · Authors · 2024-11-15
>
> __The analysis focuses only on fully trained models and does not provide insights into how ID changes over time.__
>
> Please see Fig 1 (right) for ID over training time for Pythia, the only model besides the slightly anomalous OLMo,  to release intermediate checkpoints. We find here that ID increases as a result of learning. This lends support to our findings that higher ID corresponds to learned abstractions over the input data.

---

> > ### Author Response · Authors · 2024-11-15
> >
> > __The submission wrongly mentions PCA being linear (line 061, applies only to its original form) which leads to the quick conclusion to discard it.__
> >
> > Here, we are referring to linear (standard) PCA. We included this as an example of a popular dimensionality estimation method that is familiar to readers, to then bridge into the discussion on nonlinear estimators. To clarify that we’re not talking about nonlinear variants of PCA, we can change “PCA” -> “linear PCA” in the text. Or, if the reviewer prefers, we can remove this point altogether as it’s not essential to our paper.

---

> ### Author Response · Authors · 2024-11-15
>
> __A correlation analysis to other work would add more value. My first thought was a correlation to the IB method (e.g. Tishby et al. 2000 and Schwartz-Ziv & Tishby 2017), but this may not be the only or best choice.__
>
> This is a good suggestion. Indeed the information imbalance has been shown to be an upper bound to  the transfer entropy (see Del Tatto 2024), and estimating the relative information between the representations is at the basis of the analysis framework which brought to the formulation of the IB hypothesis. However, the representations analyzed in our work are those of the last token, and the sequence of these representations is not a Markov chain (only the concatenation of the representations of all the tokens would be a Markov blanket). Therefore, it is not straightforward to compare directly our results to those discussed in the literature on the IB. We will add a sentence mentioning this topic as worthy of detailed and dedicated investigation.
>
> Nevertheless, in light of your feedback we’ve prepared an “IB-like” plot, which you can see here: https://osf.io/p5z8q?view_only=913370ebe056498791f3616fb65fbee6 . Likely due to the above differences, our plots don’t display a similar trend to the final model configuration in IB; rather they present the complementary view that, through the layers, informativity (inputs -> layer) decreases, and informativity (layer -> outputs) increases, with the ID peak marking a changepoint in processing patterns. This is in support of our current results. Please let us know whether you think this would be a good addition to the Appendix!
>
> As for comparison to other methods, Fig H.1 reproduces trends in representational similarity using linearCKA. In terms of correlation analysis of our findings to work beyond the IB and representational similarity methods, we cannot think of other papers that are sufficiently similar– this speaks to the novelty of our contribution. Let us know if you find anything we can compare to; we’d be happy to investigate!

---

> > ### Author Response · Authors · 2024-11-15
> >
> > __The first paragraph of Asset Section C.1 (lines 809-836, in particular 828-829) mentions sensitivity of ID estimation w.r.t. to noise, small scales, density variations and curvature. That analysis suggests some sort of frequency decomposition integrated with the ID estimation.__
> >
> > Gride, the algorithm we are using, is one of the few capable of estimating the ID across multiple scales efficiently and robustly, making it well-suited for analyzing natural representations.
> >
> > In general, though, yours is an excellent suggestion, which we will forward to our colleagues working specifically on the development of ID estimators. In the paper, we will discuss in more detail the criticalities related to the ID estimate, as also suggested by another reviewer. In particular, we will underline that other studies address the problem of estimating the ID on multiple scales in the presence of noise and curvature. This is a fundamentally difficult problem: as shown by the classical work of [Hein and Hudibert], a key challenge is that “for small sample sizes, it is impossible to distinguish between noise and high curvature”.
> >
> > Some works address this by using a multiscale SVD [Little et al] or identifying gaps in the spectrum of the local covariance matrix estimated on hyperspheres of increasing radii [Recanatesi et al.]. However, these spectral methods are computationally intensive, which limits their applicability to the representation of neural networks.
> >
> >
> > Recanatesi et al., A scale-dependent measure of system dimensionality \
> > Anna V. Little et al, Multiscale Geometric Methods for Data Sets I: Multiscale SVD, Noise, and Curvature \
> > Hein and Hudibert, Intrinsic Dimensionality Estimation of Submanifolds in R d

---

### Official Review · Reviewer_PyB7 · 2024-11-08

**Soundness:** 3
**Presentation:** 3
**Contribution:** 3
**Rating:** 6
**Confidence:** 3

**Summary:**

The paper explored how transformer-based language models evolve their internal representations across layers, revealing a distinct high-dimensional abstraction phase. The authors observe the some findings across multiple LMs and datasets, and they provide a foundation for better understanding and optimizing language model architectures. This work bridges the gap between geometric representation and linguistic function in transformer-based LMs. Also, it highlights the potential of intrinsic dimensionality as a tool for analyzing and evaluating LMs.

**Strengths:**

This work conducts experiments on various LMs (e.g., OPT-6.7B, Llama-3-8B, Pythia-6.9B, OLMo-7B, and Mistral-7B) using multiple datasets, providing a comprehensive analysis. It also observes how representational intrinsic dimensionality (ID) varies across layers and proposes insightful hypotheses. Furthermore, this work inspires the research community to explore the utilization of ID information in transformer-based LM applications.

**Weaknesses:**

While the paper combines two methods, GRIDE (Denti et al.) and Information Imbalance (Glielmo et al., 2020), to analyze four large language models (LLMs), this combination may fall short in terms of novelty. In Section 4.1, the choice of pre-training datasets for evaluation is also a limitation. Since these datasets have likely been encountered by the models during training, the results may not provide a fully accurate picture of the models’ generalization capabilities. Testing on unseen datasets would be crucial to evaluate the robustness and generalizability of the observed patterns, especially in real-world applications where unseen data is the norm. The study is limited to a narrow range of LLMs in terms of scale. Evaluating models of varying sizes (e.g., smaller models alongside large ones) would offer a more comprehensive understanding of how model size impacts intrinsic dimensionality and representation overlap across layers.

**Questions:**

1. There seems to have second ID peak in the later layers over LLMs. Do you think this second ID peak might reveal additional insights?
2. In your analysis (Figure 4), you observed that Pythia and OPT exhibit very similar representations. Could this similarity be attributed to pre-training on similar datasets? If so, how might this influence your findings, and have you considered controlling for dataset overlap to isolate structural factors more effectively?
3. The work focuses on classification tasks to analyze representation spaces in language models. Could you explain why generative tasks were not included? Do you expect the observed ID peaks and representation patterns to differ in generative contexts?

---

> ### Author Response · Authors · 2024-11-14
> **Response to Weaknesses**
>
> Thanks for your valuable feedback! Please find below our responses to the points raised in Weaknesses:

---

> > ### Author Response · Authors · 2024-11-14
> >
> > __GRIDE (Denti et al.) and Information Imbalance (Glielmo et al., 2020), to analyze four large language models (LLMs), this combination may fall short in terms of novelty.__
> >
> > To the best of our knowledge, our paper is the first to apply both GRIDE and Information Imbalance to studying neural network representations (and in particular those of LLMs).

---

> > ### Author Response · Authors · 2024-11-14
> >
> > __In Section 4.1, the choice of pre-training datasets for evaluation is also a limitation… Testing on unseen datasets would be crucial to evaluate the robustness and generalizability of the observed patterns, especially in real-world applications where unseen data is the norm.__
> >
> > Thanks for this point. Our question in this paper concerns how LMs process in-distribution data. This is a key feature of our setup, allowing us to make statements about the LMs’ learned behavior on the generic language they were trained on. Crucially, we do not claim that the empirical ID profiles we found generalize to out-of-distribution data. We can make this statement more clear in the Discussion.
> >
> > In addition, note that it’s difficult to tell what would be unseen data for these models. At this time, only the OLMo training data are fully publicly available. What comes closest to unseen data in our experiments could be the bookcorpus, which is a unique textual typology. Intriguingly, we observe that bookcorpus is also the dataset where ID peaks tend to be the least pronounced. This observation, together with the fact that the peaks almost disappear for shuffled data, suggests that peak ID size correlates with the degree to which processing data follow training distributional statistics.

---

> > ### Author Response · Authors · 2024-11-14
> >
> > __The study is limited to a narrow range of LLMs in terms of scale. Evaluating models of varying sizes (e.g., smaller models alongside large ones) would offer a more comprehensive understanding__
> >
> > We agree that it's important to evaluate different-sized models. This experiment is in "Influence of model size on ID and probing tasks", Appendix D, and Fig D.1, and a pointer to the Appendix section is in line 260.
> >
> > We replicated the finding that the ID peak corresponds to semantic/syntactic processing in one larger and one smaller Pythia model (Fig D.1 middle, right). The differences in ID by model size is in Fig D.1 (left). For different sizes, the ID profile over layers is correlated, but the magnitude is larger for larger models.

---

> ### Author Response · Authors · 2024-11-14
> **Responses to Questions**
>
> __There seems to have second ID peak in the later layers over LLMs. Do you think this second ID peak might reveal additional insights?__
>
> We have preliminary evidence that suggests that the second peak coincides with a phase in which the model is preparing to generate the next output token. For example, for some models performance in a surface-probing task starts increasing again under the second peak, suggesting that the model is processing actual "tokens" again. However, for space reasons, we chose to focus on the first, larger ID peak phase that is more consistent across all models.

---

> > ### Author Response · Authors · 2024-11-14
> >
> > __In your analysis (Figure 4), you observed that Pythia and OPT exhibit very similar representations. Could this similarity be attributed to pre-training on similar datasets? If so, how might this influence your findings, and have you considered controlling for dataset overlap to isolate structural factors more effectively?__
> >
> > Good question. Our work gives a robust characterization of consistent representational profiles in different models tested on different datasets. We leave to future work a _causal_ understanding of the sources of this consistency (training data, objective, architecture). As LLMs are pre-trained using large computational resources and their training data often are not public, this is not entirely trivial. Interestingly, concurrent work has shown that representations of models are converging (Huh et al, 2024) not only in language, but in other modalities as well. Our work links this convergence to the high-ID layers where linguistic abstraction happens. But, in the general case, the cause of representational similarity between different models is still unknown and is an active area of research (Moschella et al, 2023; Huh et al, 2024; see UniReps and Re-align workshops).

---

> > > ### Author Response · Authors · 2024-11-14
> > >
> > > __Could you explain why generative tasks were not included?__
> > >
> > > Another good question. Our goal with the probing tasks was to find out what kind of information about the inputs is present in LM layer activations. Classification tasks are ideal for this.
> > > A classification task on, e.g., sentiment, predicts a label from the layer representations $X_t$, ($t$ is the layer index). If sentiment classification achieves high accuracy on $X_t$, then we can say that information about sentiment is encoded in layer $t$.
> > >
> > > Why not generative tasks? Again, we want to investigate the kind of information contained in each layer. Generating text from intermediate layers has been shown to be tricky and uninterpretable (Belrose, 2023); also, it raises the question, what kind of information would need to be generated for us to interpret each layer’s function? Instead, classification via probing can be performed straightforwardly in any layer. Hope this answers your question! We will clarify this point in the final version.

---

### Official Review · Reviewer_j48a · 2024-11-09

**Soundness:** 3
**Presentation:** 3
**Contribution:** 3
**Rating:** 8
**Confidence:** 5

**Summary:**

This work takes a high-level geometric approach to analyze intrinsic dimension (ID) of the representational manifold at each layer of a decoder-only Transformer LLM to understand how layer geometry relates to layer function. Although inspired by the earlier work of (Valeriani et al., 2023), this work greatly extends the models investigated to include five mainstay decoder-only LLMs, and added more extensive probing and downstream tasks on defined datasets to analyze ID profiles across layers. The resulting observations are different from those from (Valeriani et al., 2023). This work made quite a few interesting findings on detecting broad qualitative patterns, and provides useful guidance for future research towards interpretability, analysis of model behavior and quality, and model pruning and layer-specific fine-tuning etc.

**Strengths:**

(1)	Although inspired by the earlier work of (Valeriani et al., 2023), this work greatly extends the models investigated to 5 distinct mainstay transformer-decoder-only LLMs and added more extensive probing and downstream tasks on defined datasets to analyze ID profiles across layers. Hence, the conclusions drawn in this work are verified across various models, datasets, and tasks, making the findings more convincing.

(2)	The comparisons to related works, esp. (Valeriani et al., 2023) which inspires this work, are clearly presented, hence the contributions of this work are clear and solid.
The verification of the emergence of a central high-dimensionality phase, and analysis of language processing behavior and performance during the high-dimensionality phase are quite thorough.

(3)	The analysis in Conclusion demonstrates that many findings in this work align with prior works and concurrent works. The paper clearly summarizes insights of guidance for future research. The Appendix provides detailed experimental setup and additional results. And finally, the analysis of potential applications of the findings is valuable to the research community.

(4)	Overall, the paper is clearly written and easy to follow.

**Weaknesses:**

(1)	Although the paper is overall clearly written, please make sure that every symbol used needs to be clearly defined when it first appears, e.g., d in Section 3.4.

(2)	Please provide rationale for critical algorithmic designs, for example, please clarify why GRIDE is selected, and why the three alternative measures for comparing layer’s representation spaces are chosen.

(3)	Currently, k is still selected based on visual inspection. It would be useful to propose methods that can automatically select k.

(4)	It is interesting that OLMo seems a bit of an outlier compared to the other 4 LMs, although it also exhibits the ID peak and other related properties. It would be useful to provide insights on why OLMo behaves differently from the other models, and shed light on patterns of any potential “outlier” LM.

**Questions:**

(1) Please check the questions listed under Weaknesses.

---

> ### Author Response · Authors · 2024-11-15
> **Response to Reviewer**
>
> Thanks so much for your careful review. We’re glad you found our recommendations for future work clear, and our contributions valuable to the community. We’ll respond to each comment below.

---

> > ### Author Response · Authors · 2024-11-15
> >
> > (1) Although the paper is overall clearly written, please make sure that every symbol used needs to be clearly defined when it first appears, e.g., d in Section 3.4.
> >
> > Thank you for pointing out this oversight, we are adding this in Section 3.4.

---

> > > ### Author Response · Authors · 2024-11-15
> > >
> > > __(2) Please provide rationale for critical algorithmic designs, for example, please clarify why GRIDE is selected, and why the three alternative measures for comparing layer’s representation spaces are chosen.__
> > >
> > > Thanks for the suggestion. We’ll justify our choices in the manuscript as follows:
> > >
> > > We selected GRIDE because it allows probing, in a rigorous framework, the dependence of the ID on the scale. In a complex data landscape the ID at short distances is dominated by noise, while at large distances is affected by curvature effects and density variation. GRIDE allows selecting on  a rigorous basis such an intermediate scale. It is a generalization of the TwoNN estimator (Facco et al 2017), which has enjoyed a lot of popularity in the ID estimation literature (Valeriani et al, 2023; Cheng et al, 2023; Chen et al, 2024; Doimo et al, 2024, among others). While TwoNN assumes local uniformity up to the 2nd nearest neighbor, GRIDE relaxes this assumption to produce unbiased ID estimates up to the $2^k$th nearest neighbor ($k$ being the scale).
> > >
> > > We chose to test RSA and linear CKA also because they are broadly used in the analysis of the representations of DNNs (see Sucholutsky et al, 2023; Williams, 2024 for surveys). On the other hand, we are the first to use Information Imbalance to analyze neural net embeddings. This, as we underline in the paper, allows measuring _asymmetric_ information containment (RSA/CKA are symmetric). Such information asymmetries are related to the non-linearity of the manifold, and are not captured by linear methods such as  RSA and CKA.

---

> > > > ### Author Response · Authors · 2024-11-15
> > > >
> > > > __(3) Currently, k is still selected based on visual inspection. It would be useful to propose methods that can automatically select k__.
> > > >
> > > > We selected a value of $k$ that is at the same time not too large (to retain the local nature of the estimator), and that minimizes the variation of the ID with respect to this parameter. Visually, this corresponded to the lowest $k$ for which the ID reaches a maximum. We also verified that the ID profiles were robust to the choice of this parameter, if varied around the reference value (fig C.2). We will consider automatic selection of $k$ as an important direction for future work.

---

> > > > > ### Author Response · Authors · 2024-11-15
> > > > >
> > > > > __(4) It is interesting that OLMo seems a bit of an outlier compared to the other 4 LMs, although it also exhibits the ID peak and other related properties. It would be useful to provide insights on why OLMo behaves differently from the other models, and shed light on patterns of any potential “outlier” LM.__
> > > > >
> > > > > We agree that understanding the causes for OLMo's slightly outlying behaviour should be a priority. We did not find any obvious difference in data or architecture that could easily explain it, and we plan to dedicate future work to a more causal approach to observing the emergence of ID peaks (along the lines of what we do in the paper by considering the effect of size and training data amount).

---

### Official Review · Reviewer_1CPH · 2024-11-11

**Soundness:** 3
**Presentation:** 3
**Contribution:** 2
**Rating:** 6
**Confidence:** 3

**Summary:**

The paper uses the technique of intrinsic dimension estimation as a tool for analyzing properties of different transformer LLM layers. 5 different LLMs are analyzed on textual inputs from 3 different public-domain corpora. In addition to computing the intrinsic dimensionality (ID) (using the generalized ratios intrinsic dimension estimator) for different layers, the ID is  correlated with performance of different layers' representations on syntactic and semantic probing tasks. Furthermore,  the difference in representational power between different layers is measured using an Information Imbalance criterion. The authors find that middle layers in LLMs have the highest ID; ID peaks seem to be an indicator of linguistic structure being learnt; early onset of peaks in ID across layers is correlated with better next token prediction performance performance; and high ID peak layers are representationally equivalent across different LLMs.

**Strengths:**

The paper conducts a broad analysis across 5 different LLMs and considers a range of questions and ablation studies (e.g., estimating ID on shuffled data, comparing layers across different models); altogether an impressively broad set of experiments. The paper is clearly written and presents a few new insights (e.g., correlation between peak onset and performance). Code and data would be made available, which would be valuable for the community.

**Weaknesses:**

The use of ID as an analysis tool for LLM layers is  not an entirely new idea (e.g., https://arxiv.org/pdf/2402.18048).
Most of the results (e.g., the peaking of ID at middle layers, emergence of  linguistically informative representations in those layers) has been shown before by means of other methods (e.g., mutual information or canonical correlation analysis). These should have been discussed in more detail under prior work.

**Questions:**

While the analyses show some interesting trends, it is difficult to tell how meaningful or significant the numerical differences are. Methods for analyzing LLM layers other than through ID could have been discussed in a prior work section.

---

> ### Author Response · Authors · 2024-11-14
>
> Thanks so much for your feedback, especially the references! We respond in detail to your comments below. In particular, we would greatly appreciate it if you could point us to any more missing work that you’re aware of.

---

> ### Author Response · Authors · 2024-11-14
>
> __The use of ID as an analysis tool for LLM layers is not an entirely new idea (e.g., https://arxiv.org/pdf/2402.18048).__
>
> Thanks for the reference. We were not aware of this paper and we will include it in the related work. Their emphasis is on using the local ID of specific inputs in specific layers as a truthfulness cue, which is very different from our emphasis on global ID as an indicator of general processing features of multiple LMs across layers. Like us, they also report ID profiles across layers for a specific model (confirming the overall pattern we detected), but the link we establish, for multiple LMs, between per-layer ID and properties such as linguistic processing, downstream performance and cross-layer and cross-model similarity is novel and, in our opinion, useful to understand how LLMs work.

---

> ### Author Response · Authors · 2024-11-14
>
> __Most of the results (e.g., the peaking of ID at middle layers, emergence of linguistically informative representations in those layers) has been shown before by means of other methods__
>
> We are happy to extend the related work with any relevant prior work you point out.

---

> ### Author Response · Authors · 2024-11-14
>
> __While the analyses show some interesting trends, it is difficult to tell how meaningful or significant the numerical differences are.__
>
> The trends that we show are meaningful with high statistical confidence. We have added confidence intervals based on multiple iterations (5 random data splits) of all experiments and significance scores where appropriate. We emphasize however in the paper that our interpretation of the results is essentially qualitative in nature and supported by the fact that a number of different models, datasets, experiments and measures provide converging quantitative evidence for the same high-level picture of how LMs process language. Still, we welcome ideas for further quantitative tests we'd be happy to run.

---

### Author Response · Authors · 2024-11-22
**Thanks + remaining questions**

Hi all, thanks again for your constructive reviews. As the discussion period is coming to an end, please let us know if anything remains unclear in our response.

---

### Meta-Review · Area_Chair_kRYK · 2024-12-08

**Metareview:**

The paper employs intrinsic dimension (ID) estimation as a technique to analyze the properties of different layers in transformer-based LLMs. While inspired by previous work, this study expands the scope by including five LLMs and introducing more extensive probing and downstream tasks on defined datasets to analyze ID profiles across layers. The paper presents several interesting findings. Most of the concerns raised by the reviewers were addressed during the authors' rebuttal.

**Additional Comments On Reviewer Discussion:**

Reviewer 5snc provided the lowest review score. However, the questions raised primarily concern writing issues, which can be addressed in the camera-ready version.

---

### Decision · Program_Chairs · 2025-01-22

Accept (Poster)